# Smartphone overuse, depression & anxiety in medical students during the COVID-19 pandemic

Flor M. Santander-Hernández[1], C. Ichiro Peralta[2], Miguel A. Guevara-Morales[1], Cristian Díaz-Vélez[3,4], Mario J. Valladares-Garrido[5,6]*

**1** Sociedad Científica de Estudiantes de Medicina de la Universidad Cesar Vallejo filial Piura, Universidad Cesar Vallejo filial Piura, Piura, Peru, **2** Facultad de Medicina Hipólito Unanue, Universidad Nacional Federico Villarreal, Lima, Peru, **3** School of Medicine, Universidad Privada Antenor Orrego, Trujillo, Peru, **4** Instituto de Evaluación de Tecnologías en Salud e Investigación–IETSI, EsSalud, Lima, Peru, **5** South American Center for Education and Research in Public Health, Universidad Norbert Wiener, Lima, Peru, **6** Hospital Regional Lambayeque, Chiclayo, Peru

\* mario.valladares@uwiener.edu.pe

## Abstract

### Introduction

Medical students have made particular use of smartphones during the COVID-19 pandemic. Although higher smartphone overuse has been observed, its effect on mental disorders is unclear. This study aimed to assess the association between smartphone overuse and mental disorders in Peruvian medical students during the COVID-19 pandemic.

### Methods

A cross-sectional study was conducted in 370 students aged between 16 and 41 years (median age: 20) in three universities from July to October 2020. A survey including Smartphone Dependence and Addiction Scale, PHQ-9, and GAD-7 was applied. Prevalence ratios were estimated using generalized linear models.

### Results

Smartphone overuse was a common feature among students ($n$ = 291, 79%). Depressive symptoms were present in 290 (78%) students and anxiety symptoms in 255 (69%). Adjusted for confounders, addictive/dependent smartphone use was significantly associated with presence of depressive symptoms (PR = 1.29, 95% CI: 1.20–1.38 for dependent use; PR = 1.30, 95% CI: 1.12–1.50 for addictive use). Also, addictive/dependent smartphone use was significantly associated with presence of anxiety symptoms (PR = 1.59, 95% CI: 1.14–2.23 for dependent use; PR = 1.61, 95% CI: 1.07–2.41 for addictive use).

### Conclusions

Our findings suggest that medical students exposed to smartphone overuse are vulnerable to mental disorders. Overuse may reflect an inappropriate way of finding emotional relief,

**Data Availability Statement:** All relevant data are within the manuscript and its Supporting Information files.

**Funding:** The author(s) received no specific funding for this work.

**Competing interests:** The authors have declared that no competing interests exist.

which may significantly affect quality of life and academic performance. Findings would assist faculties to establish effective measures for prevention of smartphone overuse.

## Introduction

The coronavirus disease 2019 (COVID-19) pandemic has tremendously impacted mental health and may be a source of major public health concerns in the coming years [1–5]. It has been shown that around 30–40% of people suffered from depression, anxiety, and insomnia during the pandemic [2,5–7] and these rates have significantly increased compared to previous years [2,8]. Some other studies have indicated a relationship between depression/anxiety and negative situations like family members who were diagnosed or deceased due to COVID-19, financial hardship, relationship problems, presence of insomnia, and suicidal ideation [7–11]. Furthermore, higher burden of emotional distress has led to an increase in demand for smartphones, due to social restrictions adopted by countries affected by the outbreak [12,13]. It has been shown that more than 50% of people have increased smartphone use [12,13], mostly as a way to alleviate from anxiety, isolation, and other issues [14].

The use of smartphones has increased over the years, enabling closer social communication. The Global System for Mobile Communications published a report [15] estimating that two thirds of the Latin American population has a smartphone. In Peru, according to the National Institute of Statistics [16], 93% of households have at least one smartphone with easy access to the internet. However, overuse of these devices has been linked to alarming rates of depression and anxiety [17], mainly because of distressing seeking of social reassurance [18].

It is estimated that 27% and 34% of medical students suffer from depression and anxiety worldwide [19,20]. Medical students are more likely to suffer higher rates of mental disorders than the general population [19,20], as time is spent largely on integrating concepts of basic and clinical science, putting knowledge into practice through hospital clerkships, learning to interact with patients, solving problems in life-threatening situations, and being constantly evaluated by exams and professors. The pandemic may have increased the burden of mental health problems in medical students, in particular because learning methods shifted to virtual mode, excluding many skills essential for their professional development. In this context, it is likely that smartphones were used more often for emotional support, increasing the probability of overuse even more than other student groups [13,21]. Prior research has reported a 30–40% prevalence of smartphone overuse in medical students [22,23]; however, we did not find recent reports in the current pandemic.

The relationship between smartphone overuse and mental disorders has been widely reported in university students and similar populations [24]. However, there are no consensus on which study variable represents the outcome, arguing in some cases a bidirectional association [25]. For example, Demirci et al. found that smartphone overuse was an independent predictor of high depression/anxiety scores, while Matar Boumosleh proposed a reverse order of association [26,27]. In the case of medical students, we found studies reporting only a unidirectional association: Chen et al. found anxiety and depression as independent factors for smartphone addiction [23], and Lei et al. found a positive, no adjusted linear association between depression/anxiety scores and smartphone addiction scores [22].

In addition, some studies among university students addressed the association of interest during the COVID-19 pandemic. Overall, these studies proposed a unidirectional association (depression and anxiety as predictors of smartphone overuse), but have shown conflictive results [21,28–30]. In the case of medical students, we found only a study in China that

reported an association between problematic smartphone use and anxiety using a structural equation model analysis [31]. However, to our knowledge there are no additional reports exploring the independent effect of smartphone overuse on mental disorders among medical students. The current gap needs to be addressed assuming the risk of long-term mental disorders in this population [32].

Medical students should regulate the use of smartphones as part of a healthy lifestyle, especially at a crucial time like the COVID-19 pandemic. As virtual interaction has prevailed in education and daily activities, evidence is needed in different sociocultural contexts. Our aim was to better understand how the pandemic has influenced the association between smartphone overuse and mental disorders in Peruvian medical students. Understanding this association would provide objective information that would assist faculties in targeted interventions and preventive programs to improve medical students' quality of life and academic performance.

For this purpose, we formulated the following research questions: 1) How common is presence/severity of depressive and anxiety symptoms as well as smartphone overuse in Peruvian medical students during the pandemic? 2) Are presence rates of depressive and anxiety symptoms higher in students who experience negative situations during the pandemic? 3) To what extent does smartphone overuse lead to presence of depressive and anxiety symptoms in this context? Our hypotheses were that 1) medical students would report higher presence/severity rates of depressive and anxiety symptoms, and higher rates of smartphone overuse during the COVID-19 pandemic [13,21]; 2) there would be higher presence rates of depressive and anxiety symptoms among students that experienced negative situations during the pandemic [7–11]; and 3) smartphone overuse would independently and significantly contribute to the higher presence rates of depressive and anxiety symptoms [31].

## Methods

### Study design

We conducted a cross-sectional survey study in medical students from Piura, Peru in the context of COVID-19 pandemic. Data collection occurred between July and October 2020, a period in which Peru was exposed to the first pandemic wave and established general lockdown with strict social distancing measures, such as restricted access to public spaces and limited hours of pedestrian transit. In this context, universities fully embraced e-learning platforms to deliver medical education.

Students were receiving online classes during the study period. A convenience sampling method was used. To estimate the total number of students, three authors communicated with class presidents from each university. Surveys were designed using Google forms and shared with an invitation message through common WhatsApp groups. Informed consent was displayed on the first page.

Participation was voluntary and the informed consent explained the study objective and confidential treatment of data. Information was stored in anonymized databases. After the survey, we invited all medical students from the three universities to participate in educational sessions provided by psychiatrists. These sessions were conducted as a way of thanking the participants and universities for their support of the study, and as a contribution to the prevention of smartphone overuse in medical students.

### Participants

This study initially considered the entire student population, regardless of the course the students were taking. Individuals that completed the informed consent form and responded to

the survey were included. Inclusion of participants was also based on whether they reported owning a smartphone with Internet access for use in daily activities. Exclusion occurred in individuals with a self-report of diagnosed depression, treatment with antidepressants in the last year, and age under 18 years. Students were asked if they met any of these selection criteria before starting the survey. Of these, thirty-eight were excluded on the basis of age alone. We obtained a sample of 375 students who met the selection criteria, of which five refused to participate in the study (they checked "no" on the informed consent form). The response rate was 98.6%.

The sample consisted of 370 students from three universities ($n_1 = 151$, $n_2 = 121$, $n_3 = 98$), which represented 16.6% of the population. Median age was 20 (from 16 to 41 years) and 61.9% were female. Characteristics of the sample are detailed in Table 1.

## Measures

Depressive symptoms were assessed using the Patient Health Questionnaire-9 (PHQ-9), a 9-item self-report instrument rated on a 4-point Likert scale (0 = not at all, 1 = several days, 2 = more than half the days, and 3 = nearly every day). The PHQ-9 was designed by Kroenke et al. [33] to screen for major depression according to the DSM-IV criteria. The overall scores range from 0 to 27. Symptom severity was categorized as minimal (0–4 points), mild (5–9 points), moderate (10–14 points), moderately severe (15–19 points), and severe (20–27 points). For the purpose of this study, we used a Colombian version [34] of the PHQ-9, which was validated in medical students with similar characteristics to our population. The Cronbach's alpha coefficient in the Colombian sample was 0.83 [34], while in the present study was 0.91.

**Table 1. Sample characteristics.**

| Variables | *n* | % |
|---|---|---|
| **Age*** | 20 | 16–41 |
| **Sex** | | |
| Male | 141 | 38.11 |
| Female | 229 | 61.89 |
| **Academic year** | | |
| First | 68 | 18.38 |
| Second | 72 | 19.46 |
| Third | 70 | 18.92 |
| Fourth | 68 | 18.38 |
| Fifth | 40 | 10.81 |
| Sixth | 35 | 9.46 |
| Seventh | 17 | 4.59 |
| **Marital status** | | |
| No single | 7 | 1.89 |
| Single | 363 | 98.11 |
| **Body mass index** | | |
| Low | 15 | 4.05 |
| Normal | 227 | 61.35 |
| Overweight | 100 | 27.03 |
| Obesity | 28 | 7.57 |
| **Hours of sleep**† | 6.24 | 1.61 |

*Median (min–max values).
†Mean (standard deviation).

Anxiety symptoms were assessed using the General Anxiety Disorder-7 (GAD-7) scale, a 7-item self-report instrument rated on a 4-point Likert scale (0 = not at all, 1 = several days, 2 = more than half the days, and 3 = nearly every day). The GAD-7 was designed by Spitzer et al. [35] to screen for GAD according to the DSM-IV criteria. The overall scores range from 0 to 21. Symptom severity was categorized as minimal (0–4 points), mild (5–9 points), moderate (10–14 points), and severe (15–21 points). We used a Spanish version of the GAD-7 [36] validated in the general population. The Cronbach's alpha coefficient in the Spanish sample was 0.94, while in the present study was 0.93.

The extent of smartphone use was measured with the Smartphone Dependence and Addiction Scale (SDAS), a 40-item self-report instrument rated on a 5-point Likert scale (0 = totally disagree, 1 = disagree, 2 = neither agree nor disagree, 3 = agree, and 4 = totally agree). The SDAS was designed by Aranda-López et al. [37] to assess problematic use of smartphone in specific Spanish-speaking population. The scale is divided into three components: 1) use, abuse, and addiction to smartphone and its apps (30 items); 2) personality traits (6 items); and 3) monetary expenditure on mobile apps and games (4 items). The overall scores range from 0 to 160, with higher scores indicating higher smartphone dependence. To categorize the extent of smartphone use, scores obtained in each item were averaged. Then, mean scores from each response were grouped to calculate the 25th and 75th percentile of their distribution. These percentiles were considered as cut-off points to define three levels of smartphone use: no dependent use (mean score $\leq 25^{th}$ percentile), dependent use (mean score between the $25^{th}$ and $75^{th}$ percentiles), and addictive use (mean score $\geq 75^{th}$ percentile). The Cronbach's alpha coefficient in the original sample was 0.81, 0.76, and 0.71 for components 1, 2, and 3, respectively [37]. In the present study, the Cronbach's alpha coefficient was 0.93 for the overall scale, and 0.93, 0.71, and 0.76 for components 1, 2, and 3, respectively.

The following potential confounders were included: insomnia, suicidal ideation, problems related to contagion or loss of a family member, relationship problems (defined as whether the participant has suffered a major relationship breakup during the last three months), and financial hardship (defined as whether the participant have suffered from a serious financial problem during the last 3 months). Insomnia was assessed using the Insomnia Severity Index (ISI), a 7-point self-report instrument rated on a 5-point Likert scale (from 0 = not at all, to 4 = extremely). The ISI was developed by Bastien et al. [38] to quantify perceived insomnia severity following DSM-IV criteria. The overall scores range from 0 to 28. Insomnia severity was categorized as no clinically significant (0–7 points), subthreshold (8–14 points), moderate (15–21 points), and severe (22–28 points). We used a Spanish version [39] of the ISI, which was validated in medical students and their social networks. The Cronbach's alpha coefficient in the Spanish sample was 0.82 [39], while in the present study was 0.87. Suicidal ideation was measured using the last item of the PHQ-9 ("how often have you been bothered over the past 2 weeks by thoughts that you would be better off dead, or thoughts of hurting yourself in some way?"). Suicidal ideation was positive if participants gave any response other than "not at all". The other confounders were measured using yes/no questions.

## Data analysis

Variables of interest were initially described with frequencies (*n*, %). For analysis purposes, presence of depressive/anxiety symptoms was defined as the manifestation of any level of symptom severity (from mild to severe). Therefore, the variables followed a dichotomous distribution (0 = absence, 1 = presence) to facilitate interpretation for decision-making. To statistically compare the presence rate of depressive/anxiety symptoms by covariates, bivariate analyses were performed using chi-square tests. The effect size was measured with Cramér's V

($\varphi_c$). To assess the association of smartphone overuse and presence of depressive/anxiety symptoms, generalized linear models were used with a Poisson distribution, log link function, and robust variance. Universities were considered as clusters assuming that each institution had a particular effect on the association of interest. Multivariate analysis was performed adjusting for potential confounders (age, sex, marital status, body mass index, family member diagnosed with COVID-19, family member deceased due to COVID-19, financial hardship, relationship problems, insomnia, and suicidal ideation). Prevalence ratios (PR) and 95% confidence intervals were reported. The significance level was set at 5%. Statistical analysis was performed in Stata v.16.1.

## Ethical approval

The study was approved by the research ethics committee of the Universidad Cesar Vallejo (Piura, Peru). All participants gave informed consent before continuing with the survey. We followed the ethical principles according to the Declaration of Helsinki. Data were used only for research purposes and remained confidential.

To safeguard the possible emotional distress related to some sensitive questions, we tried to carefully explain these items and clearly mentioned that the survey was anonymous and that they were free to withdraw their participation at any time during the study. In addition, we tried to emphasize that their contribution would be very important to understand the problem of excessive use of smartphones and that through them preventive measures could be provided for the benefit of students in general. Although the educational sessions were conducted to all students due to the anonymity of the surveys, we provided a link at the end of the survey with content on emotional support, prevention of negative emotional states, and contact numbers for mental health centers.

## Results

Descriptive statistics of study variables are presented in Table 2. Depressive symptoms were present in 290 (78%) students. Mild symptoms were the most prevalent ($n = 119$, 32%), followed by moderate ($n = 79$, 21%), moderately severe ($n = 55$, 15%), and severe ($n = 37$, 10%). Anxiety symptoms were present in 255 (69%) students. Mild symptoms were the most prevalent ($n = 113$, 31%), followed by moderate ($n = 82$, 22%) and severe ($n = 60$, 16%). Smartphone overuse was a common feature among students ($n = 291$, 79%), of whom the majority experienced dependent use ($n = 251$, 68%) followed by addictive use ($n = 40$, 11%).

Presence of depressive and anxiety symptoms was compared for each variable of interest, as illustrated in Table 3. Presence of depressive symptoms was moderately associated with addictive smartphone use (addictive use 100% vs. dependent use 82%, $p < 0.001$, $\varphi_c = 0.32$). Presence of anxiety symptoms was strongly associated with addictive smartphone use (addictive use 98% vs. dependent use 74%, $p < 0.001$, $\varphi_c = 0.37$).

To assess the independent association between smartphone overuse and presence of depressive and anxiety symptoms, Poisson regression analysis was performed adjusting for cluster effect within universities and potential confounders (age, sex, marital status, body mass index, family member diagnosed with COVID-19, family member deceased due to COVID-19, financial hardship, relationship problems, insomnia, and suicidal ideation). In the unadjusted analysis, addictive smartphone use (PR = 1.80, 95% CI = 1.42–2.27) and dependent smartphone use (PR = 1.47, 95% CI = 1.17–1.86) were significantly associated with presence of depressive symptoms. After model adjustment, the magnitude of association was still significant but reduced by 50% and 18%, respectively. In the unadjusted analysis, addictive smartphone use (PR = 2.49, 95% CI = 1.59–3.87) and dependent smartphone use (PR = 1.88, 95% CI = 1.35–

**Table 2. Descriptive statistics of study variables (*n* = 914).**

| Variables | *n* | % |
|---|---|---|
| **Family member diagnosed with COVID-19** | | |
| No | 146 | 39.46 |
| Yes | 224 | 60.54 |
| **Family member deceased due to COVID-19** | | |
| No | 276 | 74.59 |
| Yes | 94 | 25.41 |
| **Financial hardship** | | |
| No | 247 | 66.76 |
| Yes | 123 | 33.24 |
| **Relationship problems** | | |
| No | 298 | 80.54 |
| Yes | 72 | 19.46 |
| **Severity of insomnia** | | |
| No clinically significant | 119 | 32.16 |
| Subthreshold | 169 | 45.68 |
| Moderate | 76 | 20.54 |
| Severe | 6 | 1.62 |
| **Suicidal ideation** | | |
| No | 243 | 65.68 |
| Yes | 127 | 34.32 |
| **Level of smartphone use** | | |
| No dependent | 79 | 21.35 |
| Dependent | 251 | 67.84 |
| Addictive | 40 | 10.81 |
| **Severity of depressive symptoms** | | |
| Minimal | 80 | 21.62 |
| Mild | 119 | 32.16 |
| Moderate | 79 | 21.35 |
| Moderately severe | 55 | 14.86 |
| Severe | 37 | 10.00 |
| **Severity of anxiety symptoms** | | |
| Minimal | 115 | 31.08 |
| Mild | 113 | 30.54 |
| Moderate | 82 | 22.16 |
| Severe | 60 | 16.22 |

2.61) were significantly associated with presence of anxiety symptoms. After model adjustment, the magnitude of association was still significant but reduced by 88% and 29%, respectively. More details are shown in Table 4.

## Discussion

Prevalence of smartphone overuse was notably high (79%) in our sample of medical students (11% for addictive use and 68% for dependent use according to the SDAS). To our knowledge, this is the first Peruvian study reporting the rate of smartphone overuse in medical students during the pandemic. Similar results were reported in this context. Hosen et al. found in Bangladeshi students an 87% prevalence of problematic smartphone use [21], while Saadeh et al.

**Table 3. Presence of depressive and anxiety symptoms according to variables of interest.**

| Variables | *n* | % | χ2 | df | *p* | φ_c |
|---|---|---|---|---|---|---|
| **Presence of depressive symptoms (*n* = 290)** | | | | | | |
| **Relationship problems** | | | | | | |
| No | 223 | 74.83 | 11.36 | 1 | **0.001** | 0.18 |
| Yes | 67 | 93.06 | | | | |
| **Severity of insomnia** | | | | | | |
| No clinically significant | 60 | 50.42 | 85.94 | 3 | **<0.001** | 0.48 |
| Subthreshold | 148 | 87.57 | | | | |
| Moderate | 76 | 100.00 | | | | |
| Severe | 6 | 100.00 | | | | |
| **Suicidal ideation** | | | | | | |
| No | 164 | 67.49 | 49.53 | 1 | **<0.001** | 0.37 |
| Yes | 126 | 99.21 | | | | |
| **Level of smartphone use** | | | | | | |
| No dependent | 44 | 55.70 | 37.04 | 2 | **<0.001** | 0.32 |
| Dependent | 206 | 82.07 | | | | |
| Addictive | 40 | 100.00 | | | | |
| **Presence of anxiety symptoms (*n* = 255)** | | | | | | |
| **Relationship problems** | | | | | | |
| No | 193 | 64.77 | 12.34 | 1 | **<0.001** | 0.18 |
| Yes | 62 | 86.11 | | | | |
| **Severity of insomnia** | | | | | | |
| No clinically significant | 46 | 38.66 | 84.76 | 3 | **<0.001** | 0.48 |
| Subthreshold | 130 | 76.92 | | | | |
| Moderate | 73 | 96.05 | | | | |
| Severe | 6 | 100.00 | | | | |
| **Suicidal ideation** | | | | | | |
| No | 139 | 57.20 | 45.38 | 1 | **<0.001** | 0.35 |
| Yes | 116 | 91.34 | | | | |
| **Level of smartphone use** | | | | | | |
| No dependent | 31 | 39.24 | 50.42 | 2 | **<0.001** | 0.37 |
| Dependent | 185 | 73.71 | | | | |
| Addictive | 39 | 97.50 | | | | |

found an increased smartphone use in 85% of Jordanian university students (including medical students) [13]. Before the pandemic period, studies showed a lower but heterogeneous prevalence of smartphone overuse in medical students. Estimates were from 37% in Saudi

**Table 4. Regression results adjusted for cluster effect within universities.**

| Level of smartphone use | Presence of depressive symptoms | | | | Presence of anxiety symptoms | | | |
|---|---|---|---|---|---|---|---|---|
| | Unadjusted | | Adjusted* | | Unadjusted | | Adjusted* | |
| | PR | 95% CI | PR | 95% CI | PR | 95% CI | PR | 95% CI |
| No dependent | Ref. | | Ref. | | Ref. | | Ref. | |
| Dependent | 1.47 | 1.17–1.86 | 1.29 | 1.20–1.38 | 1.88 | 1.35–2.61 | 1.59 | 1.14–2.23 |
| Addictive | 1.80 | 1.42–2.27 | 1.30 | 1.12–1.50 | 2.49 | 1.59–3.87 | 1.61 | 1.07–2.41 |

*Adjusted for age, sex, marital status, body mass index, family member diagnosed with COVID–19, family member deceased due to COVID–19, financial hardship, relationship problems, insomnia, and suicidal ideation.

Arabia [40], 45% in India [41], 59% in Egypt [42], to 68% in Brazil [43]. Although these studies used different scales of smartphone addiction, similar frequency patterns are observed and are notoriously higher in the pandemic period. One reason for this finding is the excessive reassurance seeking provided by social media [18]. This scenario may be aggravated by the COVID-19 context, especially due to lockdown, social distance, and loss of family and friends. It is still necessary to provide more information on the prevalence of smartphone overuse during and after the pandemic outbreak, using standardized scales and validating objective cut-off values that allow for adequate comparison.

Depressive symptoms were present in 78% of participants. This result is remarkably higher than estimates found in meta-analyses during (39%, 95% CI = 29–50%) and before (27%, 95% CI: 25–30%) the pandemic period [19,32]. However, similar findings were reported in Peruvian medical students. Two studies found a high prevalence of depressive symptoms (74%) during the first pandemic wave in Peru (similar to our data collection period) [44,45]. Although PHQ-9 was used, both studies have unclear information about how education was delivered. Another study found a prevalence of 60% with DASS-21 [46], but data collection period was after the first wave and possibly online education was delivered. Another notable finding is the 37% prevalence reported during suspension of classes [47], suggesting that academic courses increase the presence rate of depressive symptoms. It must be noted that differences between estimates are related to the instrument used (e.g., PHQ-9, DASS-21, BDI), the cut-off values used to categorize scale scores, and the period in which surveys were applied. Cross-cultural features may also explain differences in the results. In particular, the young Peruvian population traditionally lives with their families for a long time, which generally strengthens emotional ties. This trait could be a protective factor in situations of personal difficulties, but also a detrimental factor when loss of family members occurs. This study adds local information on the prevalence of depressive symptoms during a specific time. Further studies should explore differences in presence/severity of depressive symptoms at other specific pandemic and post-pandemic periods.

Anxiety symptoms were present in 69% of participants. This result is higher than estimates found in meta-analyses during (47%, 95% CI = 35–59%) and before (34%, 95% CI: 29–39%) the pandemic period [20]. Prevalence of anxiety symptoms was unexpectedly higher than prior Peruvian studies (36–57%) conducted during the first wave in Peru [44–47]. However, as with depression scores, differences between estimates are related to the instrument used (e.g., GAD-7, DASS-21, BAI-21), different cut-off values, and the study period. Cross-cultural features would also explain differences in the results. Overall, medical students may suffer higher rates of anxiety due to the sensation of uncertainty, which may vary according to the impact of COVID-19 in each country. Also, distress may occur by fear of contagion, changes in teaching/learning method, and perceived helplessness in a critical situation. This study shows a high prevalence of anxiety symptoms at the initial time of the pandemic in Peru. Further research is needed to understand how anxiety levels varies in the long term.

This study is the first to assess the association between smartphone overuse on mental disorders in Peruvian medical students. Results of regression analysis suggest that the two categories of smartphone overuse (addictive and dependent use) may have an independent effect on depressive and anxiety symptoms. This association was consistent with previous studies during and before the COVID-19 outbreak. In the pre-pandemic period, a meta-analysis found a roughly 3-fold higher prevalence of depression (95% CI = 2.3–4.4) and anxiety (95% CI = 1.2–2.8) in children and young people (including university students) experiencing smartphone addiction [24]. Some other studies before and during the outbreak stated a reverse order of association [22,23,27]. To our knowledge only one study in the COVID-19 context found that smartphone overuse significantly contributed to the presence of anxiety [31]. Interestingly, a

longitudinal study in China found a notorious increase in the prevalence of depressive symptoms [48], but no change in smartphone addiction levels, suggesting that smartphone use is not influenced by the COVID-19 context. According to our results, it seems that mental disorders may arise due to smartphone overuse in medical students. This may be a consequence of a frustrated flight from emotional burden due to personal, academic, and environmental factors, a mechanism linked to reassurance seeking [18] that has intensified during the pandemic. It should be noted that some studies indicate an absent association between smartphone overuse and mental disorders [49], and propose that depressive/anxiety symptoms are predictors of addictive smartphone use [25]. Also, the influence of other variables may change the expected association, such as self-control, sleep quality, among others [50]. Our main finding is just a piece of the puzzle and further studies should explore the complex relationship of smartphone use and mental health.

The study findings give general insight into the mental health concerns of medical students during the pandemic, and addresses the problem of smartphone use as a way to worsen mental disorders. The reported association should serve as information that will help medical educators and policymakers create targeted intervention that will reduce mental health problems. In addition, we encourage the establishment of preventive programs to address smartphone overuse, helping students to adopt a healthy relationship with these devices.

This study has several limitations. First, residual confounding is present since some variables associated with mental health were not included (e.g., economic level and social support). Second, selection bias is present since sampling was not stratified by years of study. Third, the results cannot be inferred to the entire study population, as data were collected from only three universities in northern Peru. Fourth, the study design cannot establish a causal relationship between smartphone overuse and mental disorders. Fifth, data collection through online surveys may bias the results due to subjectivity in the responses. However, novel data on smartphone use was shown in Peruvian medical students. Added to the current pandemic context, this study may serve as baseline for future studies at the regional level. Also, the results are based on instruments with adequate psychometric properties, which ensures their internal validity.

## Conclusions

A significant number of medical students experience symptoms related to depression, anxiety, and smartphone overuse during the COVID-19 pandemic. Smartphone overuse represents a major source of mental disorders and may reflect an inappropriate way of finding emotional relief, which may significantly affect quality of life and academic performance. These findings would assist faculties to establish effective measures for prevention of smartphone overuse. Further research is needed to overcome the indirect and long-term effects of COVID-19 on mental health of medical students.

## Supporting information

**S1 Dataset.**
(XLSX)

**S2 Dataset.**
(XLSX)

## Acknowledgments

We thank Emanuel D. Rufino, Diego H. Arrascue-Morales, and Milagros J. Aquino-Zapata for their support in the research project.

## Author Contributions

**Conceptualization:** Mario J. Valladares-Garrido.

**Data curation:** Mario J. Valladares-Garrido.

**Formal analysis:** C. Ichiro Peralta, Mario J. Valladares-Garrido.

**Investigation:** Flor M. Santander-Hernández, C. Ichiro Peralta, Miguel A. Guevara-Morales.

**Methodology:** Flor M. Santander-Hernández, Mario J. Valladares-Garrido.

**Project administration:** Flor M. Santander-Hernández, Miguel A. Guevara-Morales, Mario J. Valladares-Garrido.

**Resources:** Mario J. Valladares-Garrido.

**Software:** Mario J. Valladares-Garrido.

**Supervision:** Cristian Díaz-Vélez, Mario J. Valladares-Garrido.

**Validation:** C. Ichiro Peralta, Cristian Díaz-Vélez.

**Visualization:** C. Ichiro Peralta, Cristian Díaz-Vélez.

**Writing – original draft:** Flor M. Santander-Hernández, C. Ichiro Peralta, Miguel A. Guevara-Morales, Mario J. Valladares-Garrido.

**Writing – review & editing:** Flor M. Santander-Hernández, C. Ichiro Peralta, Miguel A. Guevara-Morales, Cristian Díaz-Vélez, Mario J. Valladares-Garrido.

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
