## [Decision Letter · Decision Letter 0]

28 Feb 2022

PONE-D-22-01374Smartphone overuse and mental disorders in medical students during the COVID-19 pandemicPLOS ONE

Dear Dr. Valladares-Garrido,

Thank you for submitting your manuscript to PLOS ONE. After careful consideration, we feel that it has merit but does not fully meet PLOS ONE’s publication criteria as it currently stands. Therefore, we invite you to submit a revised version of the manuscript that addresses the points raised during the review process.

Reviewers have highlighted that your paper is an interesting work related to a mental health risk factors during the COVID-19 pandemic. However, reviewers 1 and 2 have pointed out some issues that should be resolved prior to publication acceptance. I agree with them, you should address their comments.

We look forward to receiving your revised manuscript.

Kind regards,

José J. López-Goñi

Academic Editor

PLOS ONE

Journal Requirements:

Additional Editor Comments:

Reviewers have highlighted that your paper is an interesting work related to a mental health risk factors during the COVID-19 pandemic. However, reviewers 1 and 2 have pointed out some issues that should be resolved prior to publication acceptance. I agree with them, you should address their comments.

Reviewers' comments:

Reviewer's Responses to Questions

**Comments to the Author**

1. Is the manuscript technically sound, and do the data support the conclusions?

Reviewer #1: Partly

Reviewer #2: Partly

2. Has the statistical analysis been performed appropriately and rigorously? 

Reviewer #1: Yes

Reviewer #2: Yes

3. Have the authors made all data underlying the findings in their manuscript fully available?

Reviewer #1: Yes

Reviewer #2: No

4. Is the manuscript presented in an intelligible fashion and written in standard English?

Reviewer #1: Yes

Reviewer #2: Yes

5. Review Comments to the Author

Reviewer #1: Overall comments

The study was conducted in three medical schools in Peru to examine the relationship between overuse of smartphones during the pandemic and mental disorders. The introduction places the study in context and the manuscript provides sufficient information regarding methods, results, and a discussion of findings. The topic itself is relevant and current and likely to be interesting to the readership of the journal. However, there are quite a few shortcomings in terms of reporting and I provided detailed feedback to each section.

Abstract:

• Please check for Grammatical errors and ensure the abstract is written in the past tense throughout. For example, the first line could be better phrased to highlight the issue.

• In describing the results beware of making claims that students had depression or had addiction. A high score on a questionnaire is only a score, to be diagnosed with depression or addiction will require more than a questionnaire. I suggest you may want to rephrase to ‘high score on the addiction …or a positive relationship ….’.

• There is too much description of results and this should be more concise with a focus on the relationship between overuse of smartphone and mental disorders

• Include an overall summary of what the results mean and why understanding this is important.

• What is missing is an overall statement of the potential contribution of findings.

Introduction:

Overall this is well-written and puts the study in context with reference to up-to-date and relevant literature, however, strengthening the rationale is likely to improve the manuscript. To this end, you may want to consider further why understanding the association between smartphone use and mental disorders in medical students is important and why this particular group of students. What is unique about medical students and in what is potential value of findings?

It will also be useful to have a clear research question at the end of the introduction and this should be driven by the literature reviewed in the introduction. Given the quantitative nature of the research, a prediction in the shape of an hypothesis should be offered, as you are already indicating a direction in your title. If you wish to be more explorative and not make a prediction than the title may need changing to ‘use’ rather than ‘overuse’.

What is missing is an overview of research to data on the topic to date. Is this the first study that examined the use of smartphone and mental disorders? In what way your research is similar or different to previous research and what is the research problem or gap it is trying to address?

Methods

Sample

• Names of the three schools are included in the manuscript and you may want t indicate whether consent was given by these institutions. Please seek guidance regarding this issue as participating organisations sometimes agree to participate on the assumption that they are not identified in any publications.

• I am intrigued by how the exclusion criteria were applied? How did you know if students had depression, received treatment etc? was it self-reported or did you have access to record? It will be useful to have some information on this point.

• There is no information regarding the sample / participants. Were all UG students invited or just from a specific year group? How many from each school took part? Sample characteristics should be provided here.

Data collection

• Providing the educational sessions on excessive use of smartphone is good practice but here you could provide a justification as to why this was done. Was it based on findings? Was it to safeguard participants?

Variables

• This section provides information on the outcome, exposure and confounding variables, but it isn’t clear why the variables were defined in such a way as no prediction was made in the introduction regarding direction.

• You have indicated established reliability of the Smartphone Dependence and addiction Scale. What about validity? Has this been established?

• Lines 95-98, you have noted how Insomnia a measured, but what about suicidal ideation, problems related to contagion or loss of a family member, relationship problems and financial problems. Were this assessed via one single question for each, or did you use an established questionnaire?

• The questions asked are emotive and may trigger a set of emotions, please indicate what measures you put in place and whether support was offered to students

Data analysis

Once a research question is included in the introduction and a research hypothesis, this will provide justification for the analysis strategy. Please ensure you clearly state which statistical tests were conducted to examine the research hypothesis.

Ethical consideration

• Please see comment regarding the names of the participating institutions within the manuscript.

• In this section you could also mention issues of safeguarding and support offer to students given the emotive nature of study.

Results

Line 114- 115: Information regarding the sample should be moved to the appropriate section in the Methods section.

Was the survey checked for reliability and validity? Cronbach alpha should have been conducted and reported before inferential statistics. Were data normally distributed? Did it meet the assumptions for further regressions?

Overall this section provides a lot of detail but unfortunately makes for quite a cumbersome read. I suggest it is revised in line with the hypothesis /es that should be included in the introduction.

There are too many long tables, all of which impact the readability of the manuscript so you may want to consider cutting down the information presented ensuring it is focused on the purpose of the study.

Discussion

• Results should not be stated again in the discussion but should be discussed more broadly and in relation to previous literature.

• In comparing results with previous study, please acknowledge potential differences and similarities that may explain the findings. i.e., comparing to China, Brazil, USA, and pre-pandemic, were the same measures used? Are the studies comparable?

• The main issue is wording and you should be careful in stating that participants had depression. As indicated above, a score on a questionnaire is just a score, an indication, it isn’t a diagnosis.

• The discussion should provide more of an interpretation of finding and what it means.

• A lot of the studies included in the discussion should be summarised in the introduction to indicate research to date in the field.

• Beware of overinterpretation of findings and provide a more concise discussion linked to the research question and hypothesis/es you will include in the introduction.

• Clearly state in what was=y your study provided some answers to a research problem, in what way it adds to the existing body of literature on the topic, and make suggestions for further research.

• Please include a discussion of the potential value of findings and how they can inform medical educators. What is the value of findings beyond just telling us that there is a relationship?

Reviewer #2: The manuscript entitled "Smartphone overuse and mental disorders in medical students during the COVID-19

pandemic" is an interesting work related to a mental health risk factors during the COVID-19 pandemic. However, many questions should be resolved before the manuscript will be considered to publication.

1. The introduction is poorly written and needs to be completed. In particular, there is too little literature on variable interests (anxiety, depression, insomnia, smartphone use, etc. during the COVID-19 pandemic. Hundreds of articles have been published since 2019, including meta-analyzes and systematic reviews. Authors should rewrite in the introduction, adding In addition, the relationship between depression / anxiety and other variables in the study was described earlier during and earlier during the COVID-19 pandemic, so this information should be presented in the introduction to make direct hypotheses about the expected associations. If the authors are interested in the prevalence rate, information should be provided in previous studies as a basis for hypotheses.

2. More information is necessary about the situation of the COVID-19 pandemic in the Peru (e.g., which wave, what the restriction levels, lockdown, e-learning or stationairy classes, etc.) during the data were collected.

3. How were the inclusion and exclusion criteria controlled? How many students have been excluded because of each criterion?

5. How many students refused from the participation in the study? What was the response rate?

6. How questionnaires were disseminated?

7. How were the educational sessions disseminated to the participants (available)? Was the survey anonymous?

8. Each questionnaire (PHQ-9, GAD-7, SDAS, ISI) should be comprehensively described, adding information about references to the original and Peruvian version of the instrument, the number of items included in each scale and subscale, response scale shoud be described in details (verbally and digitally), reliablility for each scale and subscale in the original and current sample. Also, demographic variables should be described and all categories of answer should be presented in the Method section.

9. Descriptive analysis should should be showed in the table and commented.

10. The authors stated (lines 103-104): "Student's t test was used after evaluation of normality and homoscedasticity; otherwise, we used the Mann-Whitney U test." but these results are not presented in the manuscript.

11. The results of chi-square tests shoud include more statistics (besides p-value), such as Chi-square statistic with df, and effect size (e.g., phi, Cramer's V, or Cohen's d).

12. It is unclear, how depression and anxiety in Table 2 and Table 3 was assessed (cut-off score should be described in method section, respectively).

13. Discussion should be rewritten, using additional references, which will be added in the Introduction.

6. PLOS authors have the option to publish the peer review history of their article (what does this mean?). If published, this will include your full peer review and any attached files.

Reviewer #1: No

Reviewer #2: **Yes: **Aleksandra Rogowska

---

## [Author Response · Author response to Decision Letter 0]

15 Apr 2022

Response to the editor’s and reviewer’s comments

Journal Requirements:

Response: This manuscript has been revised according to the journal style requirements (please see the revised manuscript).

Response: The study’s minimal underlying data set was uploaded as a Supporting Information file.

Response to Reviewer #1 comments:

Abstract

• Please check for Grammatical errors and ensure the abstract is written in the past tense throughout. For example, the first line could be better phrased to highlight the issue.

Response: We have checked all this sections and provided a better phrasing.

• In describing the results beware of making claims that students had depression or had addiction. A high score on a questionnaire is only a score, to be diagnosed with depression or addiction will require more than a questionnaire. I suggest you may want to rephrase to ‘high score on the addiction …or a positive relationship ….’.

Response: We have carefully revised the use of these phrases and intended to make a better description of the results.

• There is too much description of results and this should be more concise with a focus on the relationship between overuse of smartphone and mental disorders

Response: We have rewritten this section focusing on the association of interest.

• Include an overall summary of what the results mean and why understanding this is important.

Response: We included an overall summary indicating your suggestion.

• What is missing is an overall statement of the potential contribution of findings.

Response: We have added an overall statement of the potential contribution of findings.

Introduction

Overall this is well-written and puts the study in context with reference to up-to-date and relevant literature, however, strengthening the rationale is likely to improve the manuscript. To this end, you may want to consider further why understanding the association between smartphone use and mental disorders in medical students is important and why this particular group of students. What is unique about medical students and in what is potential value of findings?

Response: We added information regarding the importance of understanding the association between smartphone overuse and mental disorders in medical students, and the potential value of findings, in order to strengthen the rationale of the study.

It will also be useful to have a clear research question at the end of the introduction and this should be driven by the literature reviewed in the introduction. Given the quantitative nature of the research, a prediction in the shape of an hypothesis should be offered, as you are already indicating a direction in your title. If you wish to be more explorative and not make a prediction than the title may need changing to ‘use’ rather than ‘overuse’.

Response: Following the literature review, we added three research questions at end of the introduction. One of them is the primary question related to the association between smartphone overuse and mental disorders, while the two others are secondary and related to the description of variables of interest. Accordingly, we formulated a hypothesis for each question and offered a predictive form of the main hypothesis.

What is missing is an overview of research to data on the topic to date. Is this the first study that examined the use of smartphone and mental disorders? In what way your research is similar or different to previous research and what is the research problem or gap it is trying to address?

Response: This is not the first study that examined the association of interest. However, there is no specific information in medical students addressing the problem of smartphone overuse and mental disorders, although literature has reported data in general students and other population. This gap could be necessary to address in the context of the pandemic assuming potential long-term mental disorders in this group. Furthermore, previous research has reported an association between these variables but with no consensus on whether smartphone overuse or mental disorder is the outcome, assuming in some cases a bidirectional association. We add more information to this gap under the assumption that smartphone overuse has a negative effect on mental health.

Methods

Sample

• Names of the three schools are included in the manuscript and you may want t indicate whether consent was given by these institutions. Please seek guidance regarding this issue as participating organisations sometimes agree to participate on the assumption that they are not identified in any publications.

Response: Ethical consent was given by only one institution. We decided to deidentify the names of the three schools included in the study to ensure confidentiality of data.

• I am intrigued by how the exclusion criteria were applied? How did you know if students had depression, received treatment etc? was it self-reported or did you have access to record? It will be useful to have some information on this point.

Response: Depression and whether they received treatment were self-reported. Therefore, we excluded these participants based on their response. We clarified this in the manuscript.

• There is no information regarding the sample / participants. Were all UG students invited or just from a specific year group? How many from each school took part? Sample characteristics should be provided here.

Response: We added a line explaining that all students were invited to participate in the study. The number of students from each school were clarified.

Data collection

• Providing the educational sessions on excessive use of smartphone is good practice but here you could provide a justification as to why this was done. Was it based on findings? Was it to safeguard participants?

Response: We provided educational sessions to prevent smartphone overuse in all students from the three universities.

Variables

• This section provides information on the outcome, exposure and confounding variables, but it isn’t clear why the variables were defined in such a way as no prediction was made in the introduction regarding direction.

Response: We clarified this methodology by adding more context to the introduction and clarifying the direction of association.

• You have indicated established reliability of the Smartphone Dependence and addiction Scale. What about validity? Has this been established?

Response: Yes, it has been stablished showing adequate validity. We clarified this part the text.

• Lines 95-98, you have noted how Insomnia a measured, but what about suicidal ideation, problems related to contagion or loss of a family member, relationship problems and financial problems. Were this assessed via one single question for each, or did you use an established questionnaire?

Response: We used one single question for each of the indicted variables. For insomnia, we used the last question of the PHQ-9. The other variables were measured using a yes/no question.

• The questions asked are emotive and may trigger a set of emotions, please indicate what measures you put in place and whether support was offered to students

Response: We explained the measures we put in place to safeguard the emotional state of students. Also, we detailed the support we could give participants by providing articles about emotional support, prevention of negative emotional states, and contact numbers for mental health centers.

Data analysis

Once a research question is included in the introduction and a research hypothesis, this will provide justification for the analysis strategy. Please ensure you clearly state which statistical tests were conducted to examine the research hypothesis.

Response: We assured the research question and hypothesis in the introduction. Then, we detailed the reason of each statistical analysis and emphasized which statistical test was conducted to examine the research hypothesis.

Ethical consideration

• Please see comment regarding the names of the participating institutions within the manuscript.

Response: We detailed the name of the institution that gave authorization, but the names of the other institutions were not revealed.

• In this section you could also mention issues of safeguarding and support offer to students given the emotive nature of study.

Response: We added a paragraph mentioning issued of safeguarding and support offer to students given the emotive nature of study. We also emphasized that participants gave informed consent before continuing with the survey.

Results

Line 114- 115: Information regarding the sample should be moved to the appropriate section in the Methods section.

Response: Information from line 114-115 were moved to the participants section, along with part of their corresponding data from Table 1.

Was the survey checked for reliability and validity? Cronbach alpha should have been conducted and reported before inferential statistics. Were data normally distributed? Did it meet the assumptions for further regressions?

Response: Our survey had three questionnaires which was not validated in the present study but used previously validated instruments in similar population. Reliability of the instruments in the present study was reported using the Cronbach alpha coefficient (coefficients are presented in Measures subsection). Data regarding depression and anxiety symptoms were not evaluated for normality because we categorized the data for both outcomes as presence or absence (dichotomous response) according to the severity of symptoms. This was in line with the use of Poisson regression models, which omits the normality assumption. The outcomes met the other assumptions for this regression analysis (Y-values are counts, counts are positive integers, counts follow a Poisson distribution, explanatory variables are continuous, dichotomous, or ordinal, and observations are independent).

Overall this section provides a lot of detail but unfortunately makes for quite a cumbersome read. I suggest it is revised in line with the hypothesis /es that should be included in the introduction.

Response: The results section was revised in line with the hypothesis included in the introduction.

There are too many long tables, all of which impact the readability of the manuscript so you may want to consider cutting down the information presented ensuring it is focused on the purpose of the study.

Response: Tables were reformatted including only information relevant to the study and focused on the purpose of the study.

Discussion

• Results should not be stated again in the discussion but should be discussed more broadly and in relation to previous literature.

Response: First paragraph regarding overview of results was deleted.

• In comparing results with previous study, please acknowledge potential differences and similarities that may explain the findings. i.e., comparing to China, Brazil, USA, and pre-pandemic, were the same measures used? Are the studies comparable?

Response: We updated the literature adding information from a meta-analysis and acknowledging limitation for comparison. Also, we detailed the comparison with previous Peruvian studies conducted during the pandemic period (please revise the two paragraphs from lines 269-298).

• The main issue is wording and you should be careful in stating that participants had depression. As indicated above, a score on a questionnaire is just a score, an indication, it isn’t a diagnosis.

Response: Wording was revised, and we intended to better describe our results.

• The discussion should provide more of an interpretation of finding and what it means.

Response: We have rewritten the discussion focusing on a more detailed interpretation of findings.

• A lot of the studies included in the discussion should be summarised in the introduction to indicate research to date in the field.

Response: We summarized studies included in the discussion, as well as added some new information in both sections.

• Beware of overinterpretation of findings and provide a more concise discussion linked to the research question and hypothesis/es you will include in the introduction.

Response: We carefully revised this section and provided a more concise discussion linked to our main hypotheses.

• Clearly state in what was=y your study provided some answers to a research problem, in what way it adds to the existing body of literature on the topic, and make suggestions for further research.

Response: We stated how our study provided some answer to a research problem, the way it adds to the existing body of literature, and made suggestions for further research.

• Please include a discussion of the potential value of findings and how they can inform medical educators. What is the value of findings beyond just telling us that there is a relationship?

Response: We added a discussion of potential value of findings and how they can inform medical educators (lines 320-325).

Response to Reviewer #2 comments:

1. The introduction is poorly written and needs to be completed. In particular, there is too little literature on variable interests (anxiety, depression, insomnia, smartphone use, etc. during the COVID-19 pandemic. Hundreds of articles have been published since 2019, including meta-analyzes and systematic reviews. Authors should rewrite in the introduction, adding In addition, the relationship between depression / anxiety and other variables in the study was described earlier during and earlier during the COVID-19 pandemic, so this information should be presented in the introduction to make direct hypotheses about the expected associations. If the authors are interested in the prevalence rate, information should be provided in previous studies as a basis for hypotheses.

Response: The introduction has been revised and rewritten. In particular, we added more information on the variables of interest during the pandemic, including information from meta-analyzes. We also added a new paragraph describing the association between smartphone overuse and mental disorders (lines 76-92), using relevant studies reported before and during the pandemic. We also used this information to formulate the study hypotheses (lines 104-110).

2. More information is necessary about the situation of the COVID-19 pandemic in the Peru (e.g., which wave, what the restriction levels, lockdown, e-learning or stationairy classes, etc.) during the data were collected.

Response: More information was added about the situation of the COVID-19 pandemic in Peru (lines 114-118).

3. How were the inclusion and exclusion criteria controlled? How many students have been excluded because of each criterion?

Response: Selection criteria was controlled by self-report of participants before starting the survey. Students were asked if they met any of the criteria. 38 students were excluded because of age < 18.

5. How many students refused from the participation in the study? What was the response rate?

Response: Five participants refused to participate in the study. The response rate was 98.6%.

6. How questionnaires were disseminated?

Response: Questionnaires were disseminated through common WhatsApp groups provided by class presidents from each university.

7. How were the educational sessions disseminated to the participants (available)? Was the survey anonymous?

Response: Educational sessions were disseminated openly to all students from study universities. This was because we did not have information about the names of the students.

8. Each questionnaire (PHQ-9, GAD-7, SDAS, ISI) should be comprehensively described, adding information about references to the original and Peruvian version of the instrument, the number of items included in each scale and subscale, response scale shoud be described in details (verbally and digitally), reliablility for each scale and subscale in the original and current sample. Also, demographic variables should be described and all categories of answer should be presented in the Method section.

Response: Each questionnaire was comprehensively described adding information suggested (please revise Measures subsection).

9. Descriptive analysis should should be showed in the table and commented.

Response: Descriptive analysis was showed in the table and commented (lines 219-225).

10. The authors stated (lines 103-104): "Student's t test was used after evaluation of normality and homoscedasticity; otherwise, we used the Mann-Whitney U test." but these results are not presented in the manuscript.

Response: We initially used these tests to compare the presence of depressive/anxiety symptoms according to numeric variables such as age and hours of sleep. We have modified this part since these variables did not represent the main findings in our study.

11. The results of chi-square tests shoud include more statistics (besides p-value), such as Chi-square statistic with df, and effect size (e.g., phi, Cramer's V, or Cohen's d).

Response: We added more detail of the results of chi-square tests, including chi-square statistic, df, and effect size using Cramer's V (Table 3).

12. It is unclear, how depression and anxiety in Table 2 and Table 3 was assessed (cut-off score should be described in method section, respectively).

Response: A cutoff score for defining the presence of depressive and anxiety symptoms was described in the methods section (lines 191-194).

13. Discussion should be rewritten, using additional references, which will be added in the Introduction.

Response: We have revised and rewritten the discussion, adding references that are also part of the introduction.

---

## [Decision Letter · Decision Letter 1]

20 Jun 2022

PONE-D-22-01374R1Smartphone overuse and mental disorders in medical students during the COVID-19 pandemicPLOS ONE

Dear Dr. Valladares-Garrido,

Thank you for submitting your manuscript to PLOS ONE. After careful consideration, we feel that it has merit but does not fully meet PLOS ONE’s publication criteria as it currently stands. Therefore, we invite you to submit a revised version of the manuscript that addresses the points raised during the review process.

We look forward to receiving your revised manuscript.

Kind regards,

José J. López-Goñi

Academic Editor

PLOS ONE

Journal Requirements:

Additional Editor Comments:

The two previous reviewers were not available in this round of revision. We have secured a new reviewer, as you can see, their comments are in the previous line. Please, consider their comments carefully.

Reviewers' comments:

Reviewer's Responses to Questions

**Comments to the Author**

1. If the authors have adequately addressed your comments raised in a previous round of review and you feel that this manuscript is now acceptable for publication, you may indicate that here to bypass the “Comments to the Author” section, enter your conflict of interest statement in the “Confidential to Editor” section, and submit your "Accept" recommendation.

Reviewer #3: (No Response)

2. Is the manuscript technically sound, and do the data support the conclusions?

Reviewer #3: Yes

3. Has the statistical analysis been performed appropriately and rigorously? 

Reviewer #3: Yes

4. Have the authors made all data underlying the findings in their manuscript fully available?

Reviewer #3: Yes

5. Is the manuscript presented in an intelligible fashion and written in standard English?

Reviewer #3: Yes

6. Review Comments to the Author

Reviewer #3: The present study aims to identify the association between smartphone overuse and mental disorders in medical students during the COVID-19 pandemic. The authors observed a high frequency of smartphone overuse in medical students. In addition, the authors concluded that those students with smartphone dependence or addiction reported depression and anxiety more frequently.

This kind of research is very useful and necessary. Therefore, the proposed work can be very interesting. I wish to compliment the authors on their thoughtful work and worthwhile goal.

Overall, the article is well written, and the logic of the study is according to the goal. In addition, it is a novel study as it focuses on the evaluation of smartphone overuse of a specific population, during the COVID-19 pandemic. Even so, some considerations and suggestions are provided below.

INTRODUCTION

Although the authors have included references from previous reviewers, it is still too brief introduction. The title of the article refers to smartphone overuse and mental disorders. In the introduction, there is a short presentation of both concepts, but it is scarce. It is recommended that the authors include more information and data on the prevalence of smartphone overuse in addition to data on the prevalence of other mental disorders. The introduction talks mainly about depression, so including information on anxiety and insomnia, later evaluated in this study, will provide a greater understanding of the topic.

METHOD

Regarding the sample selection criteria, the authors are suggested to be more specific. Are all medical students included, regardless of the course they are taking? Is it an inclusion criterion that they completed the informed consent and responded to the survey? Is access to the smartphone exclusively at home or also in other places?

An online survey is used to data collection. This method has the relevant risk of subjectivity, which may bias the data. It is recommended that the author consider this aspect on future research. In addition, it should be included as a limitation of the study.

It is mentioned that following the survey, three educational sessions on the excessive use of smartphones were given. Were these sessions carried out with a specific objective and is the impact of these sessions evaluated in any way?

In this study, different questionnaires are applied to obtain the data. There is a specific questionnaire for depression and another for anxiety. Although other psychopathological variables such as insomnia or suicidal ideation are measured, both the introduction and the results deal mainly with depression and anxiety. Therefore, the authors are asked if the article should not have another title referring specifically to depression and anxiety and not to mental disorders.

RESULTS

In the results section, reference is made to relationship problems, what do the author mean by this concept? (Page 9, line 135).

DISCUSSION

The authors are encouraged to review the discussion. They are requested to be uniform when referencing citations in the text. Sometimes the reference is indicated after the punctuation mark, and other times between periods. For example: “However, a study in Korea stated that, rather than addiction, mobile phone users experience “overdependence”, a condition developing in some of them. (21) (Page 11, lines 167-168). And “A unidirectional relationship with depression was also proposed as a possible cause of smartphone addiction. (37, 38). (Page 12, lines 191-193).

In addition, the authors are suggested to review paragraph 4 of the discussion. The way in which the data are presented is confusing (page 11, lines 169-179).

With these changes, readers will be able to fully appreciate the potential clinical significance of the findings and future directions for research. I hope these proposed modifications will serve to improve the manuscript.

7. PLOS authors have the option to publish the peer review history of their article (what does this mean?). If published, this will include your full peer review and any attached files.

Reviewer #3: No

---

## [Author Response · Author response to Decision Letter 1]

2 Aug 2022

Response to Reviewer #3 comments:

Brief message to reviewer 3: We want to thank reviewer 3 for their comments. However, we have noticed that the review of the manuscript was based on the original version and not on the revised one with track changes. Despite of this, we have responded to the comments and added all the suggestions.

Introduction

1. Although the authors have included references from previous reviewers, it is still too brief introduction. The title of the article refers to smartphone overuse and mental disorders. In the introduction, there is a short presentation of both concepts, but it is scarce. It is recommended that the authors include more information and data on the prevalence of smartphone overuse in addition to data on the prevalence of other mental disorders. The introduction talks mainly about depression, so including information on anxiety and insomnia, later evaluated in this study, will provide a greater understanding of the topic.

Response: Thank you for your suggestions. We have previously revised the introduction and focused on depression & anxiety as the outcome measures in this study. We have provided prevalence data on both outcomes as well as on smartphone overuse. We have also highlighted the relationship between depression/anxiety & smartphone overuse, adding current literature on this association. Finally, we stated the research questions and study hypotheses.

Methods

2. Regarding the sample selection criteria, the authors are suggested to be more specific. Are all medical students included, regardless of the course they are taking? Is it an inclusion criterion that they completed the informed consent and responded to the survey? Is access to the smartphone exclusively at home or also in other places?

Response: Thank you. We have specified the information on selection criteria following your suggestions (lines 131-134).

3. An online survey is used to data collection. This method has the relevant risk of subjectivity, which may bias the data. It is recommended that the author consider this aspect on future research. In addition, it should be included as a limitation of the study.

Response: Thank you. We have included this limitation on lines 337-338.

4. It is mentioned that following the survey, three educational sessions on the excessive use of smartphones were given. Were these sessions carried out with a specific objective and is the impact of these sessions evaluated in any way?

Response: Thank you. In the latest version of the manuscript, we mentioned the reason for this activity. We have tried to clarify this information further (lines 127-129). Unfortunately, due to logistic limitations we could not gather data on the impact of these sessions.

5. In this study, different questionnaires are applied to obtain the data. There is a specific questionnaire for depression and another for anxiety. Although other psychopathological variables such as insomnia or suicidal ideation are measured, both the introduction and the results deal mainly with depression and anxiety. Therefore, the authors are asked if the article should not have another title referring specifically to depression and anxiety and not to mental disorders.

Response: Thank you for your feedback. We agree that the title should be more specific according to the two mental health outcomes (depression and anxiety). We have modified the title as suggested.

Results

6. In the results section, reference is made to relationship problems, what do the author mean by this concept? (Page 9, line 135).

Response: Thank you. We have added the definition of this variable to clarify its meaning (lines 181-182).

Discussion

7. The authors are encouraged to review the discussion. They are requested to be uniform when referencing citations in the text. Sometimes the reference is indicated after the punctuation mark, and other times between periods. For example: “However, a study in Korea stated that, rather than addiction, mobile phone users experience “overdependence”, a condition developing in some of them. (21) (Page 11, lines 167-168). And “A unidirectional relationship with depression was also proposed as a possible cause of smartphone addiction. (37, 38). (Page 12, lines 191-193).

Response: Thank you. In the latest version of the manuscript, we have revised all the citations following the journal requirements.

8. In addition, the authors are suggested to review paragraph 4 of the discussion. The way in which the data are presented is confusing (page 11, lines 169-179).

Response: Thank you. We have previously reviewed this paragraph. Please see paragraphs 2 and 3 of the discussion (pages 15-16, lines 275-304).

---

## [Decision Letter · Decision Letter 2]

11 Aug 2022

Smartphone overuse, depression & anxiety in medical students during the COVID-19 pandemic

PONE-D-22-01374R2

Dear Dr. Valladares-Garrido,

We’re pleased to inform you that your manuscript has been judged scientifically suitable for publication and will be formally accepted for publication once it meets all outstanding technical requirements.

Kind regards,

José J. López-Goñi

Academic Editor

PLOS ONE

Additional Editor Comments (optional):

Reviewers' comments:

Reviewer's Responses to Questions

**Comments to the Author**

1. If the authors have adequately addressed your comments raised in a previous round of review and you feel that this manuscript is now acceptable for publication, you may indicate that here to bypass the “Comments to the Author” section, enter your conflict of interest statement in the “Confidential to Editor” section, and submit your "Accept" recommendation.

Reviewer #3: All comments have been addressed

2. Is the manuscript technically sound, and do the data support the conclusions?

Reviewer #3: Yes

3. Has the statistical analysis been performed appropriately and rigorously? 

Reviewer #3: Yes

4. Have the authors made all data underlying the findings in their manuscript fully available?

Reviewer #3: Yes

5. Is the manuscript presented in an intelligible fashion and written in standard English?

Reviewer #3: Yes

6. Review Comments to the Author

Reviewer #3: After a second revision, the authors have applied most of the recommendations. The explanation that authors have given for those doubts raised are valid.

I congratulate the authors for their work and trust that the recommendations and proposed changes have been helpful.

7. PLOS authors have the option to publish the peer review history of their article (what does this mean?). If published, this will include your full peer review and any attached files.

Reviewer #3: No

---

## [Editor Report · Acceptance letter]

19 Aug 2022

PONE-D-22-01374R2 

Smartphone overuse, depression & anxiety in medical students during the COVID-19 pandemic 

Dear Dr. Valladares-Garrido:

I'm pleased to inform you that your manuscript has been deemed suitable for publication in PLOS ONE. Congratulations! Your manuscript is now with our production department. 

Kind regards, 

on behalf of

Dr. José J. López-Goñi 

Academic Editor

PLOS ONE